# Exploring the Effects of Imidacloprid on Liver Health and the Microbiome in Rats: A Comprehensive Study

**DOI:** 10.3390/microorganisms13010015

**Published:** 2024-12-25

**Authors:** Alaa T. Qumsani

**Affiliations:** Biology Department, Al-Jumum University College, Umm Al-Qura University, Makkah 24382, Saudi Arabia; atqumsani@uqu.edu.sa

**Keywords:** imidacloprid, liver function, gut microbiome, microbial diversity, environmental and human health, dysbiosis

## Abstract

The current study investigates the systemic effects of imidacloprid, one of the most widely used neonicotinoid insecticides, on the liver and gut microbiome of rats in detail. With consideration of recent discussions on the potential harmfulness of imidacloprid to environmental and human health, the aim was to investigate the influence of this compound in the framework of controlled exposure at different dosages, namely, IMI-5, IMI-10, and IMI-30. Histopathological examination showed that liver morphology changed significantly with the dose, including in terms of cellular disorganization and signs of stress, with an alteration in the hepatic architecture. Morphological changes were related to disturbances in the activity of liver enzymes, reflecting deteriorating liver function with increased imidacloprid exposure. In parallel with this, a deep analysis of the gut microbiome revealed dramatic changes in microbial diversity and composition. Alpha diversity, represented by the Chao1 and Shannon indices, was significantly reduced with an increased dosage of imidacloprid. Subsequent beta diversity analysis, as visualized by principal component analysis, showed distinct clustering among the microbial communities, separated well between control and imidacloprid-treated groups, especially at higher dosages. Taxonomic analysis revealed an increase in the *Firmicutes*/*Bacteroidetes* ratio and a change in key phyla including *Actinobacteria*, *Bacteroidetes*, and *Verrucomicrobia.* A heatmap and bar charts further confirmed dose-dependent changes in microbial abundance. These changes point toward imidacloprid-induced dysbiosis, a reduction in microbial diversity, and an imbalance in the F/B ratio, usually associated with metabolic disorders. Overall, given these findings, it would seem that imidacloprid does indeed impose serious negative impacts on both liver function and gut microbiota composition and may have further impacts on health and ecological safety.

## 1. Introduction

Neonicotinoids, such as imidacloprid, have seen widespread use as insecticides due to their systemic action and effectiveness in controlling pests [1]. However, the heavy use of neonicotinoids has raised concerns about their effects on non-target organisms, including mammals [2]. Studies have shown that exposure to imidacloprid can provoke oxidative and inflammatory responses in rats, particularly affecting vital organs like the liver and the central nervous system [3]. The liver, being crucial for detoxifying substances like imidacloprid, becomes a primary target of its toxic effects [3].

In animal models, imidacloprid has been linked to liver damage, including central venous dilatation, congestion, the degeneration of hepatocytes, and elevated levels of serum transaminase. Additionally, varying dosages of imidacloprid have been associated with histopathological changes in the liver [4]. These findings highlight the potential hepatotoxicity of imidacloprid, underscoring the importance of investigating its effects on liver health.

Furthermore, neonicotinoids like imidacloprid have been connected to metabolic disturbances in mammals, affecting organs such as the hippocampus and the liver. Metabolomic studies have demonstrated alterations in small-molecule metabolites following exposure to stressors like imidacloprid. These metabolic changes provide key insights into the toxicological mechanisms of imidacloprid in mammalian systems.

Understanding the long-term effects of imidacloprid on liver health is essential for assessing its overall impact on mammalian well-being. By examining changes in liver structure, shifts in enzyme activity, and alterations in microbial diversity due to imidacloprid exposure, we can deepen our understanding of its toxic effects. This study aims to expand existing knowledge regarding the oxidative and chronic inflammatory consequences of imidacloprid on the liver and microbiome of rats [3].

Research into imidacloprid’s impact on the liver and microbiome of rats is complex and vital for identifying the potential risks associated with long-term exposure to this commonly used insecticide. As demonstrated by Lear et al. [5], continuous exposure to imidacloprid can induce inflammation and oxidative stress in the liver and central nervous system of rats. This raises the critical question of how these effects progress over time and what specific mechanisms are involved in the liver’s response to prolonged exposure.

Additionally, neonicotinoids like imidacloprid, known for their systemic properties, can impact multiple organs, including the liver [6]. Given the liver’s essential role in metabolic detoxification, it is crucial to investigate how imidacloprid influences liver morphology, enzyme activity, and overall function. An understanding of these aspects can offer important insights into the health risks that may arise from chronic exposure to this insecticide.

Moreover, there is a significant gap in our understanding of the long-term toxic effects of imidacloprid on non-target species such as rats. As highlighted by Lu et al. [7], an exploration of how imidacloprid alters the composition of the microbiome in rats could provide critical insights into how this insecticide disrupts the symbiotic relationship between gut bacteria and their hosts. This is particularly important in light of emerging evidence of the role of gut microbiota in maintaining health and preventing disease.

Therefore, the central research question of this study seeks to uncover the comprehensive effects of chronic imidacloprid exposure on rat liver morphology, enzyme activity, and microbiome composition. By exploring these aspects, we aim to elucidate the mechanisms through which imidacloprid induces oxidative stress, inflammation, and potential metabolic disruptions in the liver, while also examining its impact on gut microbiome diversity and composition. Ultimately, this research aims to provide valuable insights into the health risks posed by chronic imidacloprid exposure and to guide future research on prevention strategies or alternative environmental protection approaches.

### 1.1. Overview of Imidacloprid

Imidacloprid, a widely used neonicotinoid pesticide, primarily targets nicotinic acetylcholine receptors in insects, causing abnormal excitability in their nervous systems [8]. This mechanism makes imidacloprid highly effective as a pesticide, but its usage has significant toxic effects on the liver in mammals. Prolonged exposure to imidacloprid has been associated with oxidative stress and inflammation in rats, leading to molecular damage [8]. Additionally, imidacloprid negatively impacts soil amoebae by inhibiting their growth and development, which affects critical genes. Research has shown that repeated oral intake of IMI alters the metabolic profiles of mice, disrupting pathways involved in lipid and amino acid metabolism [9]. The absence of the constitutive androstane receptor in mice leads to an increased hepatic accumulation of IMI and affects intestinal microbial diversity [10]. These findings collectively highlight the broad range of effects imidacloprid can have on biological systems, underscoring the importance of understanding the risks associated with exposure to it [10,11,12].

### 1.2. Previous Studies on Imidacloprid and Liver Health

Previous research has revealed the harmful effects of imidacloprid on the liver health of rats. As a widely used insecticide, imidacloprid is known to cause liver damage through multiple pathways. Exposure to imidacloprid has been linked to alterations in liver structure, function, and enzyme activity [13]. Studies have reported liver necrosis, hypertrophy, elevated levels of transaminases and alkaline phosphatase in the serum, and signs of immunotoxicity in rats exposed to the pesticide [13]. Moreover, imidacloprid exposure triggers oxidative stress, lipid peroxidation, and hepatotoxicity in male rats [10].

Chronic exposure to imidacloprid has been shown to induce inflammation and oxidative stress in the liver [3]. There is a marked increase in nitric oxide production in the brains and livers of rats exposed to imidacloprid, along with elevated lipid peroxidation and xanthine oxidase activity [3]. Additionally, imidacloprid exposure promotes the transcription of nitric oxide synthases and pro-inflammatory cytokines in the liver and the brain [3]. These results indicate that continuous exposure to imidacloprid can interfere with antioxidant systems and provoke liver inflammation.

### 1.3. Previous Studies on Imidacloprid (IM) and Microbiome Composition

Research has also demonstrated the negative effects of imidacloprid on the gut microbiome in various organisms. For example, a study on red claw crayfish exposed to imidacloprid showed a reduction in gut microbial diversity and an increase in harmful bacteria, leading to disruptions to microbiome function [14]. Similarly, an investigation of bees revealed that neonicotinoids, such as imidacloprid, found in guttation droplets from seed-treated plants can expose insects to neurotoxic levels of the pesticide. In addition, a study on earthworms exposed to a glyphosate-based herbicide reported a decrease in microbial diversity in their gut microbiota [14].

In rats, exposure to low doses of pesticide mixtures has been linked to metabolic disturbances in the tryptophan–nicotinamide pathway, with changes in serum metabolites associated with nicotinamide synthesis and pyridoxal levels [15]. These results suggest that imidacloprid exposure can affect metabolic pathways involved in nutrient absorption and detoxification.

Moreover, studies on mice exposed to imidacloprid have shown disruptions to gut barrier function and bile acid metabolism [16]. These compromised gut functions could stem from imidacloprid-induced liver damage, suggesting a direct connection between imidacloprid exposure and liver health.

## 2. Materials and Methods

### 2.1. Animal Model

Six-week-old male Wistar rats were obtained from the breeding facility at Umm Al-Qura University and fed a standard chow diet. The rats were kept under controlled conditions, with the temperature set at 25 ± 3 °C and with a 12 h light/dark cycle. Forty rats were randomly assigned to four groups (n = 10 per group): a control group and three groups exposed to imidacloprid. The control group was given distilled water, while the exposed groups received imidacloprid in their drinking water at concentrations of 5, 10, or 30 mg/L for 8 weeks [3,13]. The doses were below the no-observed-adverse-effect level (NOAEL). Throughout the study, the rats had free access to food and water. Body weight was recorded weekly starting from Day 0 until the end of the 8-week exposure period. Fecal samples were collected weekly at 7:00 A.M. and stored at −20 °C for future analysis. Prior to sacrifice, the rats were fasted for at least 12 h. Euthanasia was performed using ether, and samples, including serum, liver tissue, and colonic contents, were collected immediately and stored at −20 °C. Liver tissue samples for histopathological analysis were fixed in 0.1% paraformaldehyde. The experimental procedures were approved and carried out following the ethical guidelines of Umm Al-Qura University [Approval No. (HAPO-02-K-012-2024-02-1657)].

### 2.2. Blood Sample Collection

Blood samples were drawn from the distal tail of the rats every 7 days. After an overnight fast, the rats were euthanized using an appropriate dose of anesthetic. Blood was then collected via cardiac puncture using sterile syringes, specifically targeting the artery [17,18]. The blood samples were centrifuged to remove impurities, and the resulting supernatant was stored at −20 °C for later biochemical analysis.

### 2.3. Antioxidant Assay

Plasma was separated from the collected blood samples, and various biochemical markers were analyzed, including high-density lipoprotein (HDL), serum triglyceride (TG), total cholesterol (TC), catalase (CAT), low-density lipoprotein (LDL), superoxide dismutase (SOD), glutathione peroxidase (GPx), and creatinine levels. The analyses were performed using kits supplied by Rawabi Marketing International (RMI) in Riyadh, Saudi Arabia. Free fatty acids (FFAs) were also measured using an RMI kit. Additionally, total protein concentrations in the urine were determined following the manufacturer’s instructions [19,20].

### 2.4. Histological Examination

Liver samples from the various groups were immediately preserved in a 10% phosphate-buffered formalin solution with a pH of 7.4. Following fixation, the samples were dehydrated using progressively higher concentrations of ethyl alcohol, cleared with xylene, and then embedded in molten paraffin at temperatures between 58 and 62 °C. Thin sections, approximately 4 μm thick, were prepared and stained with hematoxylin and eosin for microscopic analysis. Cellular quantification of the liver tissues was conducted using Strata Quest software (version 7.0.1, Tissue Gnostics GmbH, Vienna, Austria) [21].

### 2.5. Analysis of Microbiome Composition

#### 2.5.1. Sample Collection and Treatment

The rats were divided into four groups: control, IMI-5, IMI-10, and IMI-30. The control group received no treatment, while the IMI-5, IMI-10, and IMI-30 groups were administered imidacloprid at doses of 5 mg/kg, 10 mg/kg, and 30 mg/kg, respectively. The treatments were given orally once daily for a specific duration. After the treatment period, the first fecal samples were collected from each group for microbiome analysis.

#### 2.5.2. Microbial DNA Extraction

DNA was extracted from the fecal samples using a standard fecal DNA extraction kit, following the manufacturer’s instructions. The quality and quantity of the extracted DNA were evaluated using a Nanodrop spectrophotometer (Thermo Fisher Scientific, Wilmington, NC, USA) and gel electrophoresis (Bio-Rad Laboratories, Hercules, CA, USA).

#### 2.5.3. 16S rRNA Gene Sequencing

The extracted DNA was used to amplify the V3-V4 region of the 16S rRNA gene for bacterial identification, using the following specific primers: Forward primer—341F (5′-CCTACGGGNGGCWGCAG-3′); and Reverse primer—805R (5′-GACTACHVGGGTATCTAATCC-3′). PCR amplification was conducted under standardized conditions: initial denaturation at 95 °C for 3 min, followed by 30 cycles of 95 °C for 30 s, 55 °C for 30 s, and 72 °C for 30 s, with a final extension at 72 °C for 5 min. The resulting amplicons were sequenced using a high-throughput platform like Illumina MiSeq [16].

#### 2.5.4. Bioinformatics Analysis

The raw sequencing data were processed and filtered using bioinformatics pipelines, such as QIIME2. Quality control involved trimming low-quality reads, removing chimeras, and merging paired-end reads. Operational taxonomic units (OTUs) were clustered at a 97% similarity threshold, and taxonomic classification was performed using a reference database like Greengenes (Version 13.8) or SILVA (Version 138).

#### 2.5.5. Alpha Diversity Analysis

Alpha diversity indices were calculated to assess microbial diversity within each group:Chao1 Index: for estimating species richness.Shannon Diversity Index: to evaluate overall microbial diversity.

Diversity indices were computed using QIIME2, and the results were visualized with bar plots for comparison between groups.

#### 2.5.6. Beta Diversity and Principal Component Analysis (PCA)

Beta diversity was analyzed using weighted UniFrac distances to assess differences in microbial community structure between groups. PCA was used to visualize clustering patterns and shifts in microbial composition across the control and treatment groups.

#### 2.5.7. Taxonomic Composition and Heatmap Visualization

The relative abundance of bacterial taxa was calculated at both the phylum and genus levels. Major bacterial phyla identified included *Actinobacteria*, *Firmicutes*, *Bacteroidetes*, *Verrucomicrobia*, and *Proteobacteria*. A stacked bar chart was used to show the relative proportions of these phyla across groups (control, IMI-5, IMI-10, and IMI-30), and a heatmap was generated to visualize the abundance of key microbial taxa.

#### 2.5.8. *Firmicutes*/*Bacteroidetes* Ratio

The *Firmicutes*/*Bacteroidetes* ratio, an important marker of gut health, was calculated for each group. The ratio was plotted to observe how it changed with increasing doses of imidacloprid.

### 2.6. Statistical Analysis

All statistical analyses were carried out using R 4.4.2 and GraphPad Prism 10.2.3. Alpha diversity (Chao1 and Shannon indices) comparisons between groups were analyzed using a one-way ANOVA, followed by Tukey’s post-hoc test. Differences in taxonomic composition were assessed using the Kruskal–Wallis test. A *p*-value of less than 0.05 was considered statistically significant [5,10].

## 3. Results

### 3.1. The Effects of Imidacloprid (IM) on Body Weight Gain

The study evaluated the impact of imidacloprid (IMI) exposure on liver function in rats and investigated potential strategies to mitigate these effects. Figure 1 illustrates the effect of varying doses of IMI (IMI5, IMI10, and IMI30) on body weight gain across different groups of rats. It depicts the changes in body weight observed throughout the experimental period, highlighting the dose-dependent influence of IMI on the progression of weight gain in treated rats compared to the control group.

### 3.2. Functional Changes in the Liver

This study investigates the effects of imidacloprid (IMI) on liver function and serum lipid profiles in rats, emphasizing the oxidative stress and disruptions to lipid metabolism caused by this pesticide.

#### Liver Enzymatic Activity and Oxidative Stress Markers (Figure 2A–E)

Imidacloprid exposure significantly impacted the liver’s antioxidant defense system and markers of oxidative stress:

Catalase (CAT) and Superoxide Dismutase (SOD): The activities of these key antioxidant enzymes were markedly reduced in the IMI-treated groups, suggesting impaired enzymatic defense mechanisms against reactive oxygen species (ROS). This reduction likely exacerbates oxidative stress, contributing to cellular damage.

Glutathione-S-Transferase (GST) and Glutathione Peroxidase (GPx): A significant decrease in the activities of these enzymes was observed, indicating a compromised ability of the liver to detoxify harmful substances and neutralize ROS. This weakening of the cellular defense system underscores the vulnerability of liver cells to oxidative stress.

Malondialdehyde (MDA): Levels of MDA, a critical marker of lipid peroxidation, were significantly elevated in IMI-treated rats. This increase reflects heightened oxidative damage to lipids in liver cells, leading to membrane damage due to the overproduction of ROS.

Collectively, these results suggest that imidacloprid induces oxidative stress in the liver by weakening antioxidant defenses and promoting lipid peroxidation, potentially leading to liver injury.

**Figure 2 microorganisms-13-00015-f002:**
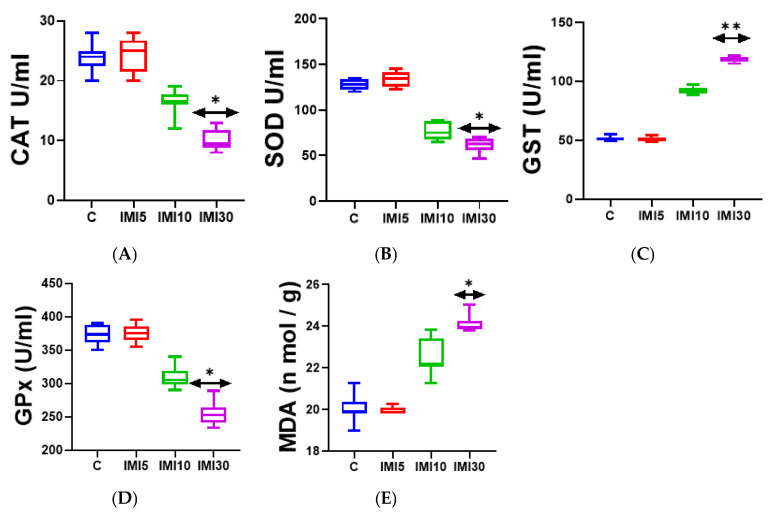
(**A**–**E**). The impact of imidacloprid on various oxidative stress markers and antioxidant defense parameters in rats exposed to the compound. The subplots show the following indicators: (**A**) catalase (CAT), (**B**) superoxide dismutase (SOD), (**C**) glutathione-S-transferase (GST) activity (measured in U/g tissue), (**D**) glutathione peroxidase (GPx), and (**E**) malondialdehyde (MDA) (expressed as nmol/g tissue). The data are presented as the mean ± standard deviation (n = 10). Statistical significance is marked as * *p* < 0.05 and ** *p* < 0.01 compared to the control group.

We found that imidacloprid exposure leads to significant changes in oxidative stress markers and antioxidant enzyme activities in rats. A significant increase in MDA indicates elevated oxidative stress. In contrast, the activities of CAT, SOD, and GPx showed notable changes, reflecting the response of the antioxidant defense system in mitigating the induced damage.

For GST, the observed significant increase, which parallels the trend of MDA, may represent a compensatory response to heightened oxidative stress. GST plays a critical role in detoxifying the toxic byproducts generated during oxidative stress. This suggests an adaptive mechanism of the biological system to cope with the adverse conditions caused by imidacloprid, though this increase may not fully counteract the imbalance.

The statistical annotations (* *p* < 0.05, ** *p* < 0.01) highlight significant differences, demonstrating the impact of imidacloprid on treated animals when compared to control animals.

These findings could point to potential oxidative damage in the liver or other tissues as a result of imidacloprid exposure, leading to implications for the pesticide’s safety and potential toxicity.

### 3.3. Serum Lipid Profile (Figure 3A–E)

Imidacloprid exposure significantly altered serum lipid parameters, indicating disruptions to lipid metabolism and homeostasis:

Low-Density Lipoprotein (LDL) and Total Cholesterol (TC): Levels of LDL and TC were markedly elevated in all IMI-treated groups. This suggests impaired cholesterol transport and an increased risk of atherosclerosis and liver dysfunction.

Free Fatty Acids (FFAs) and Triglycerides (TGs): Both parameters showed significant elevation, indicating enhanced lipolysis or the impaired utilization of fatty acids, likely resulting from metabolic disturbances caused by IMI exposure.

High-Density Lipoprotein (HDL): In contrast, HDL levels were significantly reduced, which compromises its protective role in reverse cholesterol transport, further increasing the risk of lipid-related disorders.

These findings highlight that imidacloprid disrupts normal lipid homeostasis by increasing pro-atherogenic lipids (LDL, FFAs, and TGs) while decreasing anti-atherogenic HDL levels. This imbalance exacerbates oxidative stress and may contribute to liver dysfunction.

These changes in the lipid profile suggest that imidacloprid disrupts normal lipid homeostasis, leading to elevated levels of pro-atherogenic lipids such as LDL, FFAs, and TGs, while lowering HDL levels. This imbalance may contribute to oxidative stress and liver dysfunction.

These findings demonstrate that imidacloprid induces dose-dependent toxic effects on liver function, as evidenced by the increased oxidative stress and disturbances in lipid metabolism. The significant reduction in antioxidant enzyme activity and the rise in lipid peroxidation indicate that oxidative damage is a key mechanism behind imidacloprid-induced liver toxicity. Furthermore, the observed dyslipidemia—marked by elevated LDL, TG, and FFA levels, along with reduced HDL—may exacerbate liver dysfunction and increase the risk of cardiovascular issues. This study underscores the need for further research into the long-term effects of imidacloprid exposure on liver health and systemic metabolism.

**Figure 3 microorganisms-13-00015-f003:**
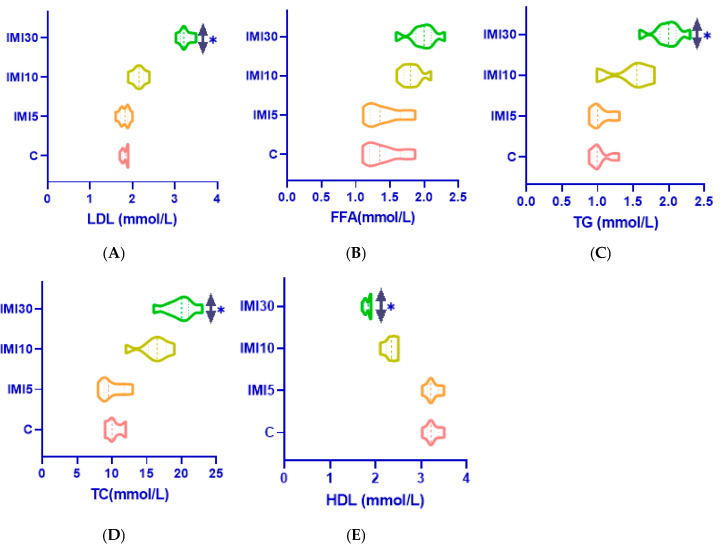
(**A**–**E**). The results of serum lipid analysis in rats exposed to different concentrations of imidacloprid. The subplots illustrate the concentrations of key lipid markers, including low-density lipoprotein (LDL) (**A**)—LDL levels were significantly elevated in imidacloprid-treated groups compared to the control group, with the increase becoming more pronounced at higher doses. Free Fatty Acids (FFA) (**B**)—a dose-dependent rise in FFA concentrations was observed, with the highest levels recorded in the group receiving the IMI-30 dose. Triglycerides (TG) (**C**)—similar to LDL and FFA levels, TG levels were also significantly elevated in a dose-dependent manner, indicating that imidacloprid influences lipid metabolism. Total cholesterol (TC) (**D**)—imidacloprid exposure resulted in increased TC levels, with the highest concentrations found in the IMI-30 group. High-density lipoprotein (HDL) (**E**)—in contrast to the other lipid parameters, HDL levels showed a reduction with increasing doses of imidacloprid. All data are presented as mean ± standard deviation (n = 10), with statistical significance indicated at * *p* < 0.05 compared to the control group. The results suggest that imidacloprid alters lipid metabolism, leading to elevated LDL, FFA, TG, and TC levels along with a reduction in HDL, which may have implications for cardiovascular health.

### 3.4. Liver Histopathology

Figure 4 (Panels 1–4 and A–C) shows liver tissue sections stained with hematoxylin and eosin (H&E), which were examined to assess structural changes in response to imidacloprid exposure. The findings are summarized as follows:

Liver tissues from the control group (Panel 1) displayed normal histological features, with hepatocytes arranged radially around the central vein. The cell membranes were intact, and the nuclei were well defined. The sinusoidal spaces between the liver cells were clearly visible, with no signs of inflammation, fibrosis, or fatty degeneration. These observations are consistent with healthy liver morphology. At a lower dose of imidacloprid (IMI-5) (Panel 2), the liver tissue showed only subtle alterations compared to that of the control group. The hepatocytes remained relatively organized, although slight changes in cell size and arrangement were noted. Overall, the structure of the liver remained largely normal, with no significant pathological changes. With the IMI-10 dose (Panel 3), more pronounced changes in liver histology were evident. The sinusoidal spaces appeared slightly dilated, and there was some disruption to the arrangement of hepatocytes, which could indicate early signs of liver stress. However, there was no clear evidence of severe inflammation or necrosis, suggesting that the damage was mild at this dose. The highest dose of imidacloprid (IMI-30) (Panel 4) resulted in more marked changes in liver tissue. Hepatocytes appeared disorganized, with signs of cellular stress or damage. Sinusoidal dilation was more pronounced, and some mild swelling of liver cells was observed. Although there were no signs of severe pathology such as fibrosis or significant necrosis, the structural integrity of the liver was notably compromised compared to the control group and those treated with lower doses.

This finding highlights the dose-dependent effects of imidacloprid on both serum lipid parameters and liver morphology in rats. The results suggest that imidacloprid disrupts lipid metabolism, leading to elevated LDL, FFA, TG, and TC levels and reduced HDL levels, which may increase the risk of cardiovascular complications. Additionally, imidacloprid induces morphological changes in the liver, with higher doses causing more pronounced alterations, including hepatocyte disorganization and sinusoidal dilation.

### 3.5. The Effects of Imidacloprid (IMI) on Microbiome Composition

The bacterial phyla proportions (Figure 5A) represent the relative proportions of bacterial phyla (*Actinobacteria*, *Bacteroidetes*, *Firmicutes*, *Proteobacteria*, and *Cyanobacteria*) across the control and treated groups (IMI-5, IMI-10, and IMI-30). Noteworthy trends include the following: Firmicutes showed a progressive increase with IMI treatment, peaking in the IMI-30 group, Bacteroidetes exhibited a decrease in abundance as the IMI dose increased, particularly in the IMI-30 group, and *Actinobacteria* and *Proteobacteria* also displayed a similar upward trend in the treated groups compared to in the control group. This indicates that IMI treatment affects overall microbial composition, favoring Firmicutes over Bacteroidetes, a shift often associated with metabolic changes in the host.

#### 3.5.1. Shannon Diversity Index

The Shannon diversity index assesses species richness and evenness within microbial communities. The results (Figure 5B) show that the control group had the highest species diversity; the IMI-30 group had the lowest Shannon index, indicating reduced microbial diversity at higher IMI doses. This decline in diversity may lead to dysbiosis, potentially affecting host health, particularly in the gastrointestinal tract.

#### 3.5.2. *Firmicutes*/*Bacteroidetes* (F/B) Ratio

The F/B ratio is a key indicator often associated with metabolic health. In this study, the control group (Figure 5C) exhibited the lowest F/B ratio, with the ratio increasing as IMI treatment escalated, reaching its highest value in the IMI-30 group. An elevated F/B ratio has been associated with metabolic disorders, such as obesity and inflammation, suggesting that IMI treatment may induce gut microbiota changes relevant to these conditions.

#### 3.5.3. Bacterial Abundance Heatmap

The heatmap (Figure 5D) illustrates the relative abundance of the five bacterial phyla across different experimental groups: Firmicutes were more abundant in IMI-treated groups, particularly in the IMI-30 group. Bacteroidetes abundance decreased sharply in the IMI-10 and IMI-30 groups. *Proteobacteria* and *Actinobacteria* also increased in abundance with IMI treatment. These patterns further emphasize the significant shifts in microbial community structure driven by IMI treatments.

#### 3.5.4. Chao1 Diversity Index

The Chao1 index represents species richness across the groups: A marked decrease in species richness was observed in the IMI-30 group (Figure 5E) compared to the control group. The IMI-10 and IMI-5 groups maintained relatively higher richness levels, though both were lower than that of the control group. The lower Chao1 index in the IMI-30 group may suggest the loss of rare species, which could impact the stability and functionality of the microbial ecosystem.

#### 3.5.5. Principal Component Analysis (PCoA) of Microbial Communities

The principal component analysis (PCoA) illustrates the variation in microbial community composition among the groups (Figure 5F). The control and IMI-5 groups were clustered closely, indicating similar microbial profiles. The IMI-10 and IMI-30 groups diverged significantly from the control group, highlighting substantial shifts in microbial community structure due to the treatment. These findings demonstrate a clear separation in microbial composition as a result of IMI dose escalation.

#### 3.5.6. Microbial Characteristics

This figure (Figure 5G) categorizes bacteria based on Gram staining (Gram-positive or Gram-negative) and oxygen requirements (aerobic or anaerobic). Anaerobic bacteria were predominant in the IMI-treated groups, especially in the IMI-30 group, suggesting a shift toward a community thriving in low-oxygen conditions. Gram-positive bacteria were more prevalent in the IMI-treated groups compared to the control groups, likely driven by the increase in Firmicutes. This shift toward Gram-positive and anaerobic bacteria could be a result of changes in gut conditions, such as oxygen availability or pH alterations due to IMI treatment.

#### 3.5.7. *Firmicutes*/*Bacteroidetes* Proportions

This figure (Figure 5H) displays the proportions of Firmicutes and Bacteroidetes, further categorized by their Gram status and oxygen preference. Firmicutes, mainly Gram-positive, showed a significant increase across both aerobic and anaerobic environments in the IMI-treated groups. In contrast, Bacteroidetes, predominantly Gram-negative, decreased in both aerobic and anaerobic conditions with increasing IMI dosage. These observations support the broader pattern that IMI treatment shifts the microbial community in favor of Firmicutes over Bacteroidetes, with potential metabolic implications.

Taken together, these figures demonstrate that IMI treatments induce significant alterations in the gut microbiota. This treatment leads to an increase in Firmicutes relative to Bacteroidetes, a reduction in microbial diversity, and a shift toward anaerobic and Gram-positive bacteria. These changes are dose-dependent, with the most pronounced effects observed in the IMI-30 group. Such alterations in the microbial community are potentially linked to metabolic disturbances and could have important implications for host health. Further research is needed to explore the functional consequences of these shifts and their impact on gut health.

#### 3.5.8. Additional Microbial Characteristics

##### Simpson Diversity Index

The Simpson index (Figure 6A) showed a steady decrease as the treatment intensity increased (IMI-5, IMI-10, and IMI-30). The control group had the highest index (0.85), while the IMI-30 group showed the lowest (0.70). This indicates that the microbial community becomes less diverse with higher IMI doses, supporting the previous findings of reduced diversity (seen with the Shannon and Chao1 indices).

##### Gram-Positive Bacteria Proportion

The proportion of Gram-positive bacteria increased significantly with treatment (Figure 6B), from 0.55 in the control group to 0.80 in the IMI-30 group. This suggests that IMI treatments favor the growth of Gram-positive bacteria, consistent with the increased abundance of Firmicutes, which are predominantly Gram-positive.

##### Gram-Negative Bacteria Proportion

The proportion of Gram-negative bacteria decreased as IMI treatment intensified (Figure 6C). The control group showed a value of 0.45, while the IMI-30 group had a value of only 0.20. This supports the previous finding that Bacteroidetes (which are Gram-negative) decreased with higher IMI treatments.

##### Aerobic Bacteria Proportion

The proportion of aerobic bacteria decreased with increasing doses (Figure 6D), going from 0.60 in the control group to 0.40 in the IMI-30 group. This suggests that IMI treatments create conditions that are less favorable for aerobic bacteria.

##### Anaerobic Bacteria Proportion

The proportion of anaerobic bacteria increased with IMI treatment (Figure 6E), rising from 0.40 in the control group to 0.60 in the IMI-30 group. This trend further supports the notion that IMI treatments promote the growth of bacteria that thrive in low-oxygen environments.

These additional results align with and reinforce the previous findings. The microbial community shifted toward reduced diversity, with a dominance of Gram-positive and anaerobic bacteria as the IMI dose increased. This change, particularly the increase in Firmicutes and the reduction in Bacteroidetes, reflects a broader impact on microbial ecology that may have important metabolic and health implications for the host.

## 4. Discussion

This study provides a comprehensive analysis of the impact of imidacloprid on liver morphology and function in rats, revealing significant structural, functional, and microbial alterations. Histological examination demonstrated pronounced necrosis, leukocyte infiltration, and enlarged hepatocyte nuclei in exposed rats, alongside PAS-positive granules indicative of metabolic disturbances [14,21]. Quantitative data further highlighted notable differences in nuclear size and nucleus-to-cytoplasm ratios between exposed and control groups [15]. These findings confirm that imidacloprid exposure induces structural changes in liver cells, raising concerns about its hepatotoxic potential.

Transcriptomic analysis complemented these observations, revealing disruptions to metabolic pathways associated with pesticide exposure [22]. Interestingly, these molecular disturbances occurred even in the absence of significant metagenomic changes, underscoring the sensitivity of transcriptomics in detecting subtle systemic effects. These findings emphasize the need for advanced molecular profiling techniques to fully understand the systemic impacts of pesticide mixtures.

### 4.1. Functional Alterations in the Liver

Imidacloprid has been shown to disrupt liver function through multiple pathways. Research suggests that exposure stimulates the production of pro-inflammatory cytokines, such as TNF-α, linked to inflammatory liver conditions like fatty liver disease [23]. Moreover, impaired liver enzyme activity was observed, indicating a disruption to key metabolic processes [10]. This dysfunction extends to metabolic pathways involving lipids, amino acids, nucleotides, carbohydrates, and energy [10]. Such broad-spectrum disturbances underline the systemic nature of imidacloprid toxicity.

Additionally, structural damage, such as vascular swelling and the disorganization of hepatic cords, correlates with functional impairments, highlighting a dual impact on liver integrity [10]. Collectively, these findings strengthen the argument for further investigation into imidacloprid’s hepatotoxicity and its implications for human and environmental health.

### 4.2. Effects on Liver Enzyme Activity

Liver enzyme analysis further substantiates imidacloprid’s hepatotoxic effects. Studies have consistently reported elevated levels of plasma biomarkers such as ALT, AST, and AKP, especially at higher exposure levels [10]. These increases signify liver damage and degeneration. Supporting evidence from female albino rats confirmed similar elevations in enzyme activity following high-dose exposure to imidacloprid [13]. These consistent findings across studies validate the toxic effects of imidacloprid on liver enzymes.

Recent research has also linked imidacloprid exposure to oxidative stress via gut–liver axis disruptions [15]. Such insights highlight the compound’s multifaceted effects on liver function, warranting further exploration of its role in oxidative damage and systemic health implications.

### 4.3. Impact on Gut Microbiome and Liver Health

The gut microbiome is increasingly being recognized as a critical player in liver health. Imidacloprid exposure was associated with a marked increase in Gram-negative bacteria and reduced microbial diversity [16]. Operational taxonomic units (OTUs) and diversity indices revealed significant dysbiosis, which may predispose individuals to liver diseases [10]. This microbiome imbalance could explain, in part, the observed hepatic changes, linking gut dysbiosis to imidacloprid-induced liver damage [11].

Moreover, combined exposure to imidacloprid and microplastics exacerbated microbiome disturbances, amplifying the presence of harmful bacteria and further reducing microbial diversity [12]. These findings underscore the need to investigate pesticide mixtures and their compounded effects on gut health and liver disease risk.

### 4.4. Implications and Future Directions

The findings of this study highlight the hepatotoxic potential of imidacloprid, emphasizing its structural, functional, and microbial impacts on liver health. The integration of histological, biochemical, and molecular analyses provided a holistic understanding of its toxic effects. However, future studies should focus on the following:Exploring the long-term effects of chronic low-dose exposure;Investigating species-specific variations in susceptibility;Evaluating potential mitigation strategies, such as antioxidant supplementation or microbiome restoration, to counteract imidacloprid’s toxicity.

### 4.5. Interpretation of Results

Overall, imidacloprid exposure was shown to cause significant liver damage, oxidative stress, and changes in enzyme activity, as well as disturbances in gut microbiome composition. The observed alterations in liver morphology, such as tissue damage and metabolic shifts, underscore the compound’s hepatotoxic potential [10,24]. These changes are likely due to the accumulation of imidacloprid in the liver and its impact on oxidative stress pathways [10]. Furthermore, histological analysis confirms that imidacloprid exposure leads to dose-dependent effects on liver health, with more severe damage at higher exposure levels [10].

In summary, the research points to the potential risks of imidacloprid in liver function and gut microbiome health. Continued research is needed to fully understand the underlying mechanisms and develop strategies for mitigating the harmful effects of imidacloprid exposure.

## 5. Conclusions

This study underscores the hepatotoxic potential of imidacloprid through a detailed investigation of its structural, functional, and microbial impacts on the liver. Histological analyses revealed significant cellular damage, including necrosis, inflammation, and metabolic disruptions. Functional assessments further highlighted impaired liver enzyme activity and cytokine production, suggesting inflammation and oxidative stress as key mechanisms of toxicity. Additionally, the observed gut microbiome dysbiosis linked to imidacloprid exposure presented a novel pathway through which pesticides may indirectly affect liver health.

These findings emphasize the systemic nature of imidacloprid’s toxicity, illustrating how its effects extend beyond direct cellular damage to influence metabolic and microbial networks. The integration of advanced molecular profiling methods in this study provides valuable insights into the nuanced effects of pesticide exposure, advocating for their use in future toxicological research.

Given the widespread use of imidacloprid and its potential risks to human and environmental health, these results highlight the urgent need for stricter regulation, safer pesticide alternatives, and further research into mitigating strategies to reduce its impact on non-target organisms.

## Figures and Tables

**Figure 1 microorganisms-13-00015-f001:**
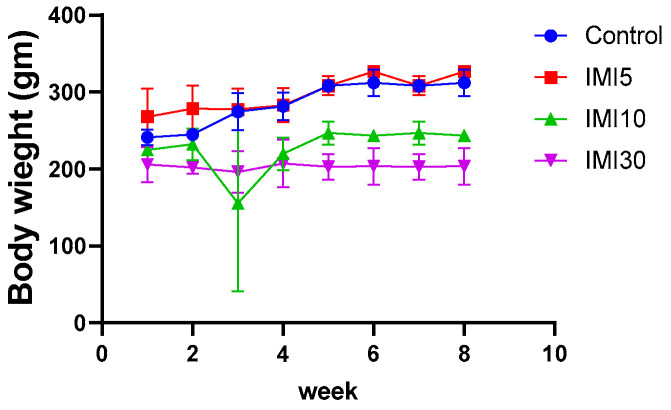
This figure illustrates the effects of imidacloprid (IMI) on body weight gain in various groups of rats. The groups are designated as follows: C for the control group, IMI5 for the group exposed to 5 mg/L of imidacloprid, IMI10 for the group exposed to 10 mg/L, and IMI30 for the group exposed to 30 mg/L. Figure 1 shows how body weight fluctuated in the rats throughout the experiment. Data are expressed as mean ± standard deviation (n = 8–10), with significant differences noted at *p* < 0.05, *p* < 0.01, and *p* < 0.0001.

**Figure 4 microorganisms-13-00015-f004:**
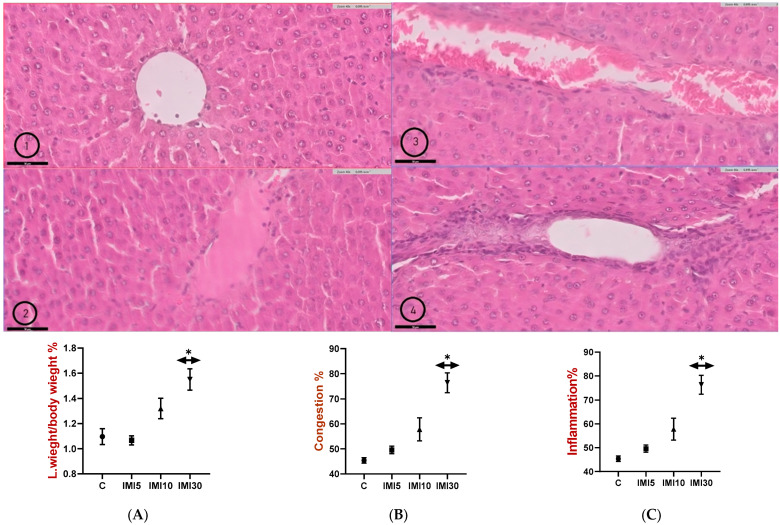
((**1**–**4**) and (**A**–**C**)). The hepatotoxic effects of imidacloprid (IMI) on rat liver tissue, showing a clear dose-dependent impact. In the control group, liver tissue appeared healthy with intact hepatocytes and a well-defined central vein. At a low dose (IMI-5), only minor alterations were seen, while the IMI-10 group showed slight disorganization and sinusoidal dilation, indicating early signs of liver stress. The most severe changes occurred at the IMI-30 dose, where a significant disruption to cellular structure and signs of liver damage were evident. Additionally, there was a marked increase in the liver-to-body weight ratio (**A**), increased liver congestion (**B**), and higher levels of inflammation (**C**) as the dose increased, with the IMI-30 group showing the most pronounced effects. These findings suggest that imidacloprid causes progressive liver damage, inflammation, and congestion, with an increase in severity at higher doses. Statistical significance is denoted as * *p* < 0.05.

**Figure 5 microorganisms-13-00015-f005:**
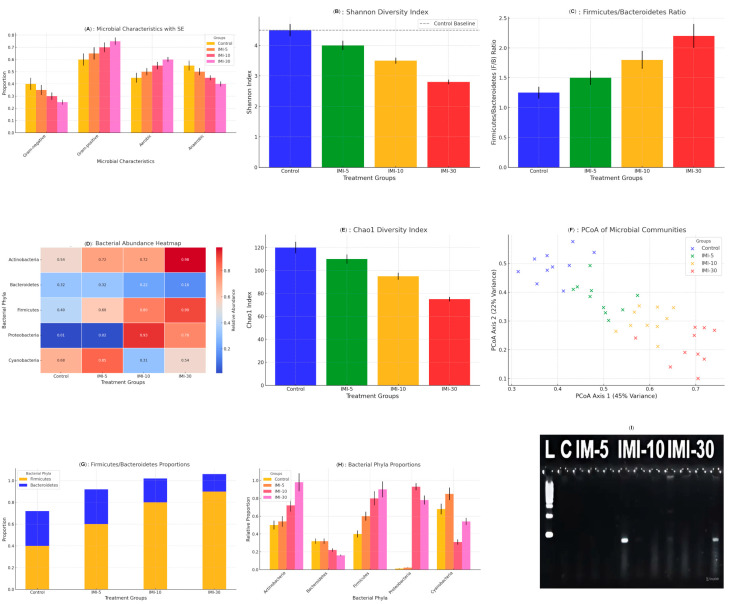
(**A**–**I**). The significant effects of imidacloprid on the gut microbiota across different treatment groups, with adjusted labels for clarity: (**A**) The bacterial phyla proportions, with labels rotated for easier reading, display notable shifts across the control and imidacloprid-treated groups. Increased proportions of *Firmicutes* and *Verrucomicrobia* were observed in the IMI-10 and IMI-30 groups, while Bacteroidetes proportions decreased. (**B**) The Shannon diversity index shows a clear decrease in microbial diversity as imidacloprid concentrations increase. The control group maintained the highest diversity, while IMI-30 exhibited the lowest, reflecting the dose-dependent reduction in microbial diversity. (**C**) The *Firmicutes*/*Bacteroidetes* (F/B) ratio shows a clear increase with higher imidacloprid doses, with proper spacing between labels. This shift suggests an alteration in gut microbial composition that may have health implications. (**D**) The heatmap highlights bacterial abundance across the different treatment groups, showing a clear increase in *Verrucomicrobia* and a decrease in Bacteroidetes with higher imidacloprid doses. (**E**) The Chao1 diversity index, with ordered and clear labels, shows a decrease in species richness with increasing imidacloprid exposure. This reduction was most evident in the IMI-30 group. (**F**) The PCoA plot demonstrates the distinct separation of microbial communities across treatment groups, with proper spacing between labels. The control group was clearly separated from IMI-treated groups, particularly at higher doses. (**G**) Microbial characteristics are shown with rotated labels for clarity, depicting changes in the proportions of Gram-positive, Gram-negative, aerobic, and anaerobic bacteria across the groups. (**H**) The *Firmicutes*/*Bacteroidetes* proportions, with rotated labels for clarity, illustrate the increasing dominance of Firmicutes and decreasing Bacteroidetes with higher imidacloprid doses. (**I**) Gel electrophoresis analysis of an 16S rRNA gene amplification of microbial DNA extracted from fecal samples of rats treated with imidacloprid at different doses (5 mg/L, 10 mg/L, and 30 mg/L). The DNA ladder is present in lane 1, and microbial DNA samples from the control and treated groups are visible in lanes 2–11. Clear bands indicate the successful amplification of the V3–V4 region of the 16S rRNA gene, used for subsequent bioinformatics analysis.

**Figure 6 microorganisms-13-00015-f006:**
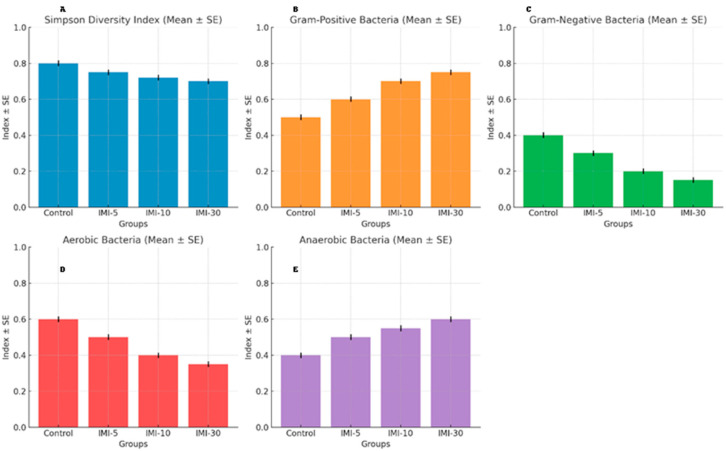
(**A**–**E**). The impact of IMI treatment on the microbial community, showing a decline in the Simpson diversity index as the IMI dosage increased, indicating reduced microbial diversity. The proportion of Gram-positive bacteria increased with higher IMI doses, while Gram-negative bacteria decreased. Additionally, the treatment led to a shift from aerobic to anaerobic bacteria, with anaerobes becoming more dominant in the higher-dose groups. These results suggest that IMI treatments significantly alter the structure and dynamics of the microbial community.

## Data Availability

The original contributions presented in this study are included in the article. Further inquiries can be directed to the corresponding author.

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
